# Impact of the COVID-19 Lockdown on Inhaler Adherence in Patients with COPD: A South Korean Nationwide Cohort Study

**DOI:** 10.3390/healthcare13121431

**Published:** 2025-06-15

**Authors:** Hyungmin Kim, Hyunduck Kim, Yookyung Yoon, Song Hee Hong

**Affiliations:** 1College of Pharmacy, Seoul National University, Seoul 08826, Republic of Korea; yanki78@naver.com; 2National Health Insurance Service, Wonju 26464, Republic of Korea; minky2030@naver.com (H.K.); jjangyunkr@empas.com (Y.Y.); 3Research Institute of Pharmaceutical Sciences, Seoul National University, Seoul 08826, Republic of Korea

**Keywords:** access, COPD (chronic obstructive pulmonary disease), COVID-19 lockdown, medication access, PDC (proportion of days covered), retrospective cohort study

## Abstract

**Background/Objectives:** The outbreak of coronavirus disease 2019 (COVID-19) has restricted access to healthcare, increasing the risk of poor disease control among patients with COPD (Chronic Obstructive Pulmonary Disease). This study aimed to compare adherence to inhalers in patients with COPD before and during the COVID-19 lockdown and determine the characteristics of patients who were adherent to inhaler medications. **Method:** A retrospective analysis was conducted on a cohort of patients with severe COPD aged 40 or older using South Korea’s National Health Insurance Service (NHIS) database, which documents all healthcare utilization covered for insured individuals. Medication adherence, measured by the proportion of days covered (PDC), was compared before and during the COVID-19 lockdown using a paired t-test. A multivariable logistic regression model was conducted to identify the characteristics of the adherent patients (socio-demographic, including age, sex, income level, insurance type, and residential area), health-conditions (disease severity, underlying diseases, and disability), and pharmacotherapy characteristics (prescriber practice setting, polypharmacy, medication treatment duration, and inhaler type). **Result:** A total of 15,971 COPD patients were identified (79.2% men). During the COVID-19 lockdown in 2020, there was a significant decrease in medication adherence to COPD inhalers compared to 2019 (49.8% in 2020 vs. 56.3% in 2019, respectively; *p* < 0.001). Moreover, the proportion of those adherent (≥80%) during the COVID-19 lockdown decreased (22.0% → 18.0%). Patients who remained adherent to inhaler therapy during the COVID-19 lockdown were typically aged in their 60s, beneficiaries of Medical Aid, residents of rural areas, clinic users, and patients without cardiovascular disease. **Conclusions:** Patients with COPD encountered significant challenges in accessing inhalers during the COVID-19 lockdown. Healthcare authorities should develop targeted strategies to ensure continued medication access for patients at increased risk of poor medication adherence, particularly during periods of restricted healthcare access, such as public health emergencies or pandemic lockdowns.

## 1. Introduction

Chronic obstructive pulmonary disease (COPD) is a progressive lung disease characterized by persistent airflow limitation and respiratory symptoms. It is a major global health burden, ranking as the fourth leading cause of death worldwide, with an estimated 3.5 million deaths attributed to the disease in 2021 [1]. Beyond its health impact, COPD imposes a substantial economic burden, with healthcare costs reaching approximately 38.6 billion Euros in Europe alone [2]. Effective management of COPD relies on long-term adherence to inhaled medications, which have been shown to reduce exacerbations, hospitalizations, and healthcare costs [3,4]. Conversely, poor adherence is associated with increased mortality, higher hospitalization rates, and elevated healthcare expenditures [3,4,5].

The outbreak of coronavirus disease 2019 (COVID-19) led to a global public health crisis, significantly disrupting healthcare access due to lockdown measures and social distancing policies. In South Korea, the government implemented strict public health interventions to mitigate the spread of COVID-19, including restrictions on in-person healthcare visits and mandatory screening before hospital access [6]. Access to tertiary hospitals was strictly regulated through mandatory COVID-19 screening and infection control protocols, making it more difficult for patients with chronic diseases, including COPD, to seek in-person medical care. Primary care institutions also had to implement pre-screening measures and remote consultation options, further altering healthcare accessibility for COPD patients [7,8]. Although these measures were necessary for pandemic control, they may have inadvertently disrupted routine disease management, potentially affecting adherence to inhaler therapy. Particularly for vulnerable populations, such as elderly patients, unfamiliarity with remote consultations compared to traditional face-to-face visits may have adversely affected healthcare accessibility.

Therefore, during the COVID-19 lockdown, there was a significant decrease in healthcare utilization, such as emergency department visits, medication adherence, and rehabilitation service [9,10]. Although most studies have shown that medication adherence among patients with chronic diseases decreased during the COVID-19 lockdown period [11,12], studies on COPD inhaler adherence have reported conflicting results, showing either an increase or decrease [13,14]. Given that COPD patients are particularly vulnerable to severe COVID-19 complications, it is plausible that some patients prioritized medication adherence to mitigate infection risks [15]. However, restricted healthcare access may have simultaneously contributed to non-adherence. Thus, understanding how inhaler adherence was affected during the COVID-19 lockdown is essential for informing healthcare policies and intervention strategies during future public health emergencies.

Medication adherence in COPD is influenced by socio-demographic factors, health conditions, and pharmacotherapy-related factors. Socio-demographic factors, such as age and income level, play a relatively important role in adherence. Similarly, health conditions, including disease severity and underlying disease, can affect patient compliance. Additionally, pharmacotherapy characteristics, such as the prescriber practice setting, polypharmacy, medication treatment duration, and inhaler type, contribute to adherence behaviors [2,16,17]. A comprehensive understanding of these factors is necessary to develop targeted interventions aimed at improving inhaler adherence, particularly during healthcare disruptions such as the COVID-19 pandemic.

This study aimed to assess changes in inhaler adherence among COPD patients before and during the COVID-19 lockdown in South Korea. Additionally, we sought to identify the factors associated with high adherence. The findings from this study will provide valuable insights for healthcare policymakers in designing effective intervention strategies to ensure continuity of care for COPD patients during future healthcare crises.

## 2. Materials and Methods

### 2.1. Study Design

A retrospective cohort study was designed to compare inhaler adherence among patients with COPD during the one-year periods before and during the COVID-19 lockdown, using South Korea’s National Health Insurance Service (NHIS) database (Figure 1). The NHIS database contains comprehensive records of healthcare utilization, including sociodemographic characteristics, medical diagnoses, and service claims for all insured residents, making it a valuable resource for nationwide population-based research.

In January 2020, South Korea implemented social distancing measures in response to the COVID-19 outbreak. These measures aimed to prevent unnecessary visits to healthcare institutions in order to preserve treatment capacity while strengthening infection control efforts, such as testing, contact tracing, and restrictions on gatherings [6,7,8,18]. The patient cohort consisted of individuals diagnosed with COPD in 2018 who were prescribed inhaler therapy, enabling a comparison of medication adherence between the periods before and during the COVID-19 lockdown (Figure 1). The pre-lockdown period was defined as calendar year 2019, the year immediately preceding the pandemic-related restrictions, whereas the lockdown period was defined as calendar year 2020, corresponding to the implementation of nationwide social distancing measures. Accordingly, we compared data from 2019 and 2020 to assess changes in inhaler adherence associated with the COVID-19 lockdown.

This retrospective cohort study was approved by the Seoul National University Institutional Review Board (IRB No. E2406/002-012).

### 2.2. Study Population, Inculsion Criteria, and Exclusion Criteria

The inclusion criteria for this study were patients aged ≥40 years who were diagnosed with COPD (ICD code: J40, J41, J42, J43, J44) in 2018 and were prescribed COPD inhalers (inhaled corticosteroids with long-acting β2-agonists (ICS_LABA) or inhaled long-acting β2-agonists with long-acting muscarinic antagonists (LABA_LAMA)) for at least 180 days during that year (Figure 2, Appendix A). According to the COPD guideline, ICS_LABA and LABA_LAMA are typically prescribed to patients classified as Group D—those with severe disease characterized by a modified Medical Research Council (mMRC) dyspnea score of ≥2 and a history of at least two exacerbations or one or more COPD-related hospitalizations [2]. The inclusion criterion of ≥180 days of inhaler prescriptions in 2018 was based on previous studies reporting approximately 50% adherence among patients with COPD, allowing for the identification of individuals with sufficient baseline exposure to maintenance inhaler therapy [3,19]. Additionally, exclusion criteria were applied to patients who had died in 2018 or 2019 and those who were no longer eligible for National Health Insurance coverage during the observation period, including cases of nationality loss or emigration.

### 2.3. Measurement of Variables

#### 2.3.1. Medication Adherence

Medication adherence was assessed using the proportion of days covered (PDC), a widely accepted metric that estimates adherence by calculating the percentage of days a patient has access to the prescribed medication over a defined observation period [20,21]. In this study, PDC was calculated as the number of days on which COPD inhalers were prescribed divided by the total number of days in the calendar year. For patients who died during the observation period, the denominator was adjusted to reflect the number of days from January 1st of that year up to the date of death. High adherence is commonly defined as a proportion of days covered (PDC) of 80% or higher. This threshold is widely accepted in adherence research and has been utilized in various studies involving chronic disease medications [21,22,23]. Therefore, we classified a PDC of 80% as high adherence.

#### 2.3.2. Socio-Demographic Variables

Socio-demographic variables included age, sex, income level, insurance type, and residential area. Age was categorized into five groups: 40–49, 50–59, 60–69, 70–79, and 80 years or older. Insurance type was classified as either national health insurance (NHI) or Medical Aid. Residential area was grouped into urban and rural categories. These variables were measured at the time of the initial ICS_LABA or LABA_LAMA prescription in 2018.

#### 2.3.3. Health Conditions

Health conditions were identified based on comorbidities requiring clinical attention and concomitant treatment, as outlined in the COPD treatment guidelines. These included cardiovascular disease, diabetes, musculoskeletal disorders, mood disorders, and lung cancer [2,24,25]. Disease severity was defined by the occurrence of acute exacerbation events, characterized by a worsening of respiratory symptoms requiring hospitalization or additional medical intervention, such as the use of quick-acting bronchodilators and oral corticosteroids [24]. All conditions were assessed using International Classification of Diseases, 10th Revision (ICD-10) codes recorded from healthcare visits during the 2018 calendar year (Appendix A). Disability status was determined based on the presence of a registered disability grade, as identified in the prescription records for the year 2018.

#### 2.3.4. Pharmacotherapy Characteristics

Pharmacotherapy characteristics included prescriber practice setting, polypharmacy, medication treatment duration, and inhaler type. The prescriber practice setting was categorized as either a clinic or a hospital, based on the type of healthcare institution most frequently visited by the patient for COPD-related care in 2018. Polypharmacy was assessed by counting the number of distinct drug classes (active pharmaceutical ingredients) prescribed during the 2018 calendar year. Medication treatment duration was determined by identifying the year of the patient’s first ICS_LABA or LABA_LAMA prescription. The initiation year was grouped into four categories: 2015, 2016, 2017, and 2018, with 2015 representing the first year that these inhalers became available through the national drug reimbursement formulary. Inhaler type was classified as either ICS_LABA or LABA_LAMA, based on the predominant number of days of use recorded in 2018.

#### 2.3.5. Sample Size Computation

We conducted a statistical power analysis to estimate the minimum sample size required for a logistic regression model including 30 covariates, aimed at detecting differences in medication adherence between two groups with markedly unequal proportions (90% vs. 10%). This scenario was chosen to reflect the most conservative case for sample size estimation. Based on these assumptions, a total sample size of at least 8692 participants would be necessary to achieve 80% power at a two-sided significance level of 0.05. Our study cohort (n = 15,971) exceeded this minimum requirement, indicating sufficient power to detect statistically significant associations in the multivariable analysis.

#### 2.3.6. Data Analysis

First, we assessed the change in medication adherence before and during the COVID-19 lockdown using a paired *t*-test among the target study population. Baseline characteristics were summarized based on the proportion of days covered (PDC) before and during the lockdown. Categorical variables were presented as frequencies and percentages and were compared using chi-square tests.

We also compared baseline characteristics between patients with high and low medication adherence during the lockdown period using chi-square tests. To identify factors associated with high adherence (PDC ≥80%) during the COVID-19 lockdown, we performed a multivariable logistic regression analysis. The results were presented as odds ratios (ORs) with 95% confidence intervals (CIs), and a *p*-value of <0.05 was considered statistically significant. All statistical analyses were conducted using SAS Enterprise Guide software (version 8.2; SAS Institute Inc., Cary, NC, USA).

## 3. Results

### 3.1. Baseline Characteristics of the Study Cohort

A total of 15,971 patients with COPD were included in the final analysis. The majority were male (79.2%), and the largest age group was patients in their 70s (40.5%). Urban residents accounted for 57.4% of the cohort. With respect to comorbidities, musculoskeletal disorders were the most prevalent (28.1%), followed by cardiovascular disease (16.8%). Lung cancer was the least common (1.8%) (Table 1, Appendix A).

Regarding pharmacotherapy characteristics, most patients initiated inhaler treatment in 2016 (37.9%). Patients were slightly more likely to be prescribed an ICS_LABA (55.3%) than a LABA_LAMA (44.7%) (Table 1). On average, patients were prescribed 1.37 distinct medications per prescription (Table 1).

### 3.2. Comparison of Inhaler Adherence Before and During the COVID-19 Lockdown

The average proportion of days covered (PDC) significantly declined from 56.3% in 2019 (pre-lockdown) to 49.8% in 2020 (during the lockdown) (*p* < 0.0001). This trend of reduced adherence was consistently observed across nearly all demographic and clinical subgroups, with statistically significant differences noted (Table 1). The proportion of patients classified with high adherence (PDC ≥80%) declined from 22.0% before the lockdown to 18.0% during the lockdown (Table 2). As shown in Figure 3, 9.1% of patients (n = 1457) transitioned from high to low adherence, whereas only 5.2% (n = 830) improved from low to high adherence, indicating a net decline in adherence status.

### 3.3. Comparison Between Patients with High and Low Adherence During the Lockdown

Significant differences were observed across all categories of baseline characteristics between patients with high and low adherence during the COVID-19 lockdown (Table 2, Appendix A). Patients in their 60s, those covered by Medical Aid, and those living in rural areas were more likely to demonstrate high adherence. In contrast, low adherence was more common among those with cardiovascular or musculoskeletal comorbidities.

In terms of pharmacotherapy-related factors, high adherence was more frequently observed among patients who visited clinics rather than hospitals, those prescribed fewer medications per prescription, those with a longer duration of inhaler use (initiated before 2018), and those using ICS/LABA therapy.

### 3.4. Determinants of High Medication Adherence During the COVID-19 Lockdown

Multivariable logistic regression analysis identified several factors significantly associated with high adherence during the lockdown (Figure 4). The strongest predictor was prior high adherence before the lockdown (OR = 18.71; 95% CI, 16.95–20.64; *p* < 0.0001).

Other significant predictors of high adherence included being in the 60–69 age group (OR = 1.19; 95% CI, 0.87–1.64), receiving Medical Aid (OR = 1.32; 95% CI, 1.09–1.59), residing in rural areas (OR = 1.17; 95% CI, 1.06–1.30), and primarily receiving care in clinics (OR = 1.42; 95% CI, 1.29–1.58). Conversely, having cardiovascular disease was associated with lower odds of high adherence (OR = 0.87; 95% CI, 0.75–0.99) (Figure 4).

## 4. Discussion

This study found that patients with COPD had significantly lower adherence to inhaler medications (ICS/LABA or LABA/LAMA) during the COVID-19 lockdown compared to the year preceding the lockdown. Moreover, the decline was consistent across all category groups, suggesting a widespread impact of the pandemic on medication behaviors among patients with COPD. The findings contrast with some previous studies [14,26] and align with others [13], likely due to differences in study settings, healthcare systems, or pandemic responses.

Reduced access to routine physician visits appears to have been a primary driver of this decline [8]. Patients with COPD, particularly older adults, may have avoided medical facilities out of fear of COVID-19 exposure, given their increased vulnerability to severe or fatal outcomes [11]. This fear may have been exacerbated by the crowding of clinics during mass vaccination campaigns, which heightened the perceived risk of infection.

In addition, changes in healthcare delivery likely contributed to reduced adherence. Hospitals, especially large tertiary institutions, including designated anchor hospitals in each regional district, were required to reallocate resources to prioritize the treatment of high-risk COVID-19 patients [8,11,27]. As a result, chronic disease management, including COPD care, may have been deprioritized, further discouraging patients from seeking routine care visits and thus contributing to decreased adherence to inhaler therapy.

Notably, older adults, patients with cardiovascular disease (CVD), those receiving care primarily at hospitals, and urban residents exhibited lower adherence during the lockdown. Older COPD patients and those with CVD likely avoided healthcare visits due to perceived infection risks. This pattern is in line with previous studies reporting similar declines in medication adherence among high-risk groups during the pandemic [28,29]. Those visiting hospitals may have experienced more limited access, as these facilities were subject to strict screening protocols and prioritized acute care [8]. Interestingly, urban residents demonstrated lower adherence than rural residents, despite generally better healthcare infrastructure. This may reflect stricter lockdown enforcement and greater reliance on in-person care in urban settings. In contrast, rural patients who typically face longer travel distances may have benefited more from the remote consultation services introduced during the pandemic [30].

Our study highlights the critical importance of managing chronic conditions such as COPD during public health emergencies, such as the COVID-19 pandemic. Notably, respiratory infections, such as COVID-19, can exacerbate COPD symptoms, thereby increasing the need for timely and consistent clinical management. However, our findings indicate a significant decline in inhaler adherence during the COVID-19 lockdown, likely driven by restricted access to healthcare services and patients’ fear of infection. This underscores the need for proactive healthcare strategies to ensure continuity of care and medication adherence among vulnerable populations during crisis situations.

In response to the COVID-19 crisis, the South Korean government adopted a national strategy based on transparency and openness, implementing the 3T approach—Testing, Tracing, and Treatment—to control the spread of infection and stabilize the situation [31]. Specifically, South Korea utilized bio-information–communication technology (BICT) to implement preemptive testing and contact tracing, enabling the early isolation of infected individuals. As a result, the country exhibited a shorter time to peak (TTP) and a lower peak height (PH) in its epidemic curve compared to other countries. This rapid and proactive policy response ultimately contributed to a lower case fatality rate [18]. Additionally, the government implemented remote physician consultations and social support services targeting vulnerable populations, including older adults and individuals with disabilities [32]. Although the overall effectiveness of these interventions remains unclear, they may have been particularly advantageous for rural residents by reducing the need for travel to healthcare facilities [7,31,32].

Supporting systems appeared to play an important role in maintaining adherence among economically vulnerable populations. South Korea operates a universal National Health Insurance (NHI) system, alongside a tax-funded Medical Aid program for individuals below the poverty line [33]. The latter provides subsidies for premiums and out-of-pocket costs. Our findings show that patients receiving Medical Aid have higher medication adherence than those covered by NHI, contradicting earlier studies [34]. It is possible that during the lockdown, Medical Aid played a stronger protective role by alleviating financial and access barriers, thereby helping patients maintain adherence.

We also observed that ICS_LABA users had significantly higher medication adherence compared to LABA_LAMA users. This finding may be attributed to the fact that ICS_LABA was covered by the national health insurance earlier than LABA_LAMA in South Korea. As a result, ICS_LABA users tended to have an earlier initiation year of treatment, which may have influenced their medication adherence (ratio of medication history [2016, 2015]: ICS_LABA 45.3% vs. LABA_LAMA 40.4%).

Although not statistically significant, lower adherence was also observed among patients with polypharmacy and those with a shorter duration of inhaler use. Polypharmacy—an increasingly prevalent issue among older adults with multiple comorbidities—may reduce adherence by increasing regimen complexity and medication burden [35]. Similarly, patients with limited inhaler experience may have lacked sufficient familiarity with device use or awareness of their disease severity, both of which can contribute to suboptimal adherence.

Finally, this study found a pronounced gender disparity among patients with COPD, with males accounting for 79.2% and females 20.8% of the study population. This finding aligns with previous studies on COPD prevalence in South Korea [36,37]. The observed imbalance is likely attributable to historically higher smoking rates among men, a well-established risk factor for COPD.

### Policy Implications

These findings highlight the need for more resilient healthcare policies to ensure continuity of chronic disease management during public health emergencies. Although remote physician consultations were introduced in South Korea, other effective strategies, such as 90-day medication supplies, home delivery services, and drive-through pharmacy pickup, were not integrated into the public healthcare system [38]. Expanding access to these services could help mitigate treatment disruptions during future crises.

In addition, our findings underscore the importance of managing chronic conditions, such as COPD, primarily within the primary care setting, where continuity and accessibility of care are more feasible. The observation that patients who primarily received care in hospital settings demonstrated lower adherence further supports this recommendation, suggesting that hospital-based care may be less conducive to effective long-term disease management for chronic conditions.

The Second National Health Insurance Comprehensive Plan, launched recently in 2024, presents an opportunity to address these challenges. The plan includes initiatives to strengthen the primary care management of chronic diseases and improve polypharmacy oversight in collaboration with community-based care systems [39]. Moreover, the National Health Insurance Service (NHIS) is currently implementing case management strategies to promote appropriate medication use among patients on multiple prescriptions [40].

The results of this study provide timely and relevant evidence to guide these policy efforts. By identifying patient-level and system-level factors influencing medication adherence during a period of healthcare disruption, this study can inform strategies to ensure the rational use of medications and protect the health of vulnerable populations during future public health emergencies. Moreover, this study offers valuable evidence not only for policymakers aiming to strengthen pandemic response frameworks, but also for the broader community by emphasizing the critical importance of maintaining continuous healthcare engagement during times of crisis.

## 5. Limitations

This study has several limitations that should be considered when interpreting the findings. First, the analysis was based on claims data, which do not capture actual medication use or patient behaviors, such as proper inhaler technique. Therefore, adherence was inferred from prescription refill records, which may overestimate or underestimate true adherence. Second, clinical variables, such as spirometry results, symptom severity, blood eosinophil counts, and CAT (COPD Assessment test) scores, were not available in the dataset, limiting our ability to stratify patients based on disease characteristics or guideline-specific treatment indications. Third, this study did not account for all possible factors influencing adherence, including family members or caregiver support, digital literacy, or access to private telemedicine services, which may have varied across individuals and regions. Fourth, although the findings provide valuable insights within the context of South Korea’s healthcare system, generalizability to other countries with different healthcare infrastructures and pandemic responses may be limited.

### Future Research

Given the inherent limitations of this retrospective study, a prospective design is warranted to enable a more precise evaluation of medication adherence among patients with COPD. Future research incorporating validated, multidimensional adherence assessment tools, such as those developed under the ENABLE COST project (COST Action: European Network to Advance Best Practices & Technology on Medication Adherence), would strengthen the evidence base on inhaler adherence. These tools would also allow for the examination of social and caregiving factors, including the potential role of family members or caregivers in preventing medication non-adherence.

In addition, the notable gender imbalance observed in our cohort highlights the need for gender-stratified analyses in future studies. Such analyses may reveal differential patterns in adherence behavior between male and female patients, thereby improving the generalizability and relevance of findings in COPD medication management.

## 6. Conclusions

This study found that medication adherence to inhaler therapy among patients with COPD significantly declined during the COVID-19 lockdown, with consistent reductions observed across socio-demographic groups. Factors such as restricted healthcare access, fear of infection, resource reallocation within hospitals, and individual characteristics, including comorbidities and treatment duration, affected adherence outcomes. Interestingly, certain support systems, such as Medical Aid and remote consultation services, may have mitigated access barriers for vulnerable populations, including rural residents. These findings underscore the importance of strengthening primary care and community-based services, especially during public health emergencies. Disruptions in routine care during public health crises can result in decreased medication adherence, the delayed management of chronic conditions, and a heightened burden on the healthcare system. To mitigate these risks, national health care systems should invest in **patient-centered primary care infrastructure** capable of maintaining continuity of care and supporting vulnerable populations during emergencies. As South Korea advances its Second National Health Insurance Comprehensive Plan, the findings of this study provide timely and policy-relevant evidence to inform strategies aimed at promoting medication adherence and safeguarding the continuity of care for patients with chronic diseases such as COPD.

## Figures and Tables

**Figure 1 healthcare-13-01431-f001:**
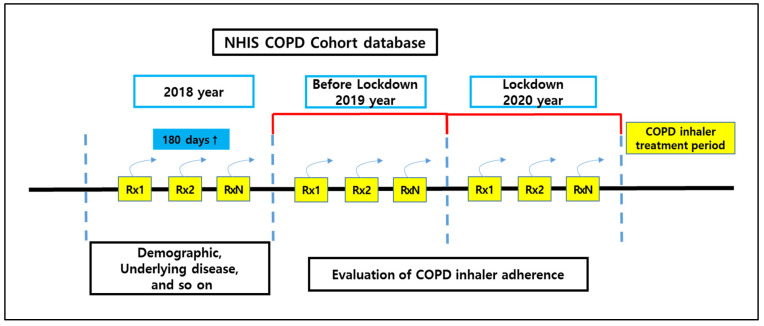
Nationwide cohort of patients with COPD before and during the COVID-19 lockdown. Note: NHIS, National Health Insurance Service; COPD, chronic obstructive pulmonary disease; Rx, Prescription.

**Figure 2 healthcare-13-01431-f002:**
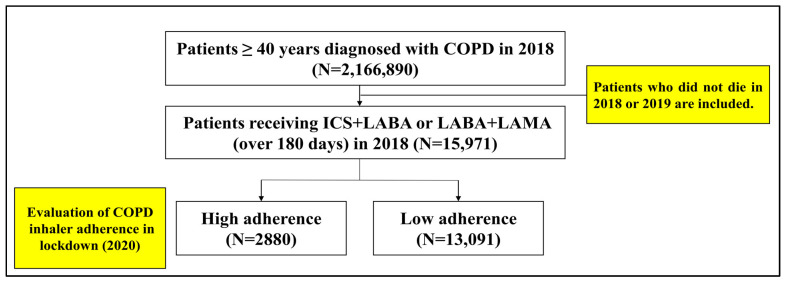
Flowchart of patient selection for the COPD inhaler adherence cohort study. Note: COPD, chronic obstructive pulmonary disease; ICS+LABA, inhaled corticosteroids with long-acting β2-agonists; LABA+LAMA, long-acting β2-agonists with long-acting muscarinic antagonists.

**Figure 3 healthcare-13-01431-f003:**
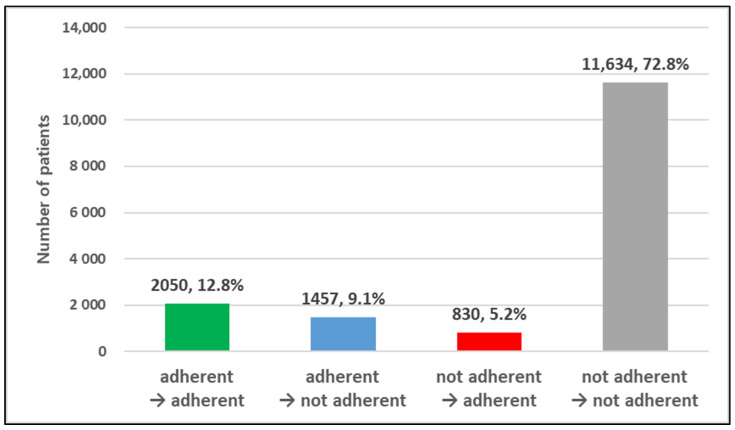
Changes in medication adherence from before to during the COVID-19 lockdown. Note: adherent if PDC (proportion of days covered) ≥ 80%.

**Figure 4 healthcare-13-01431-f004:**
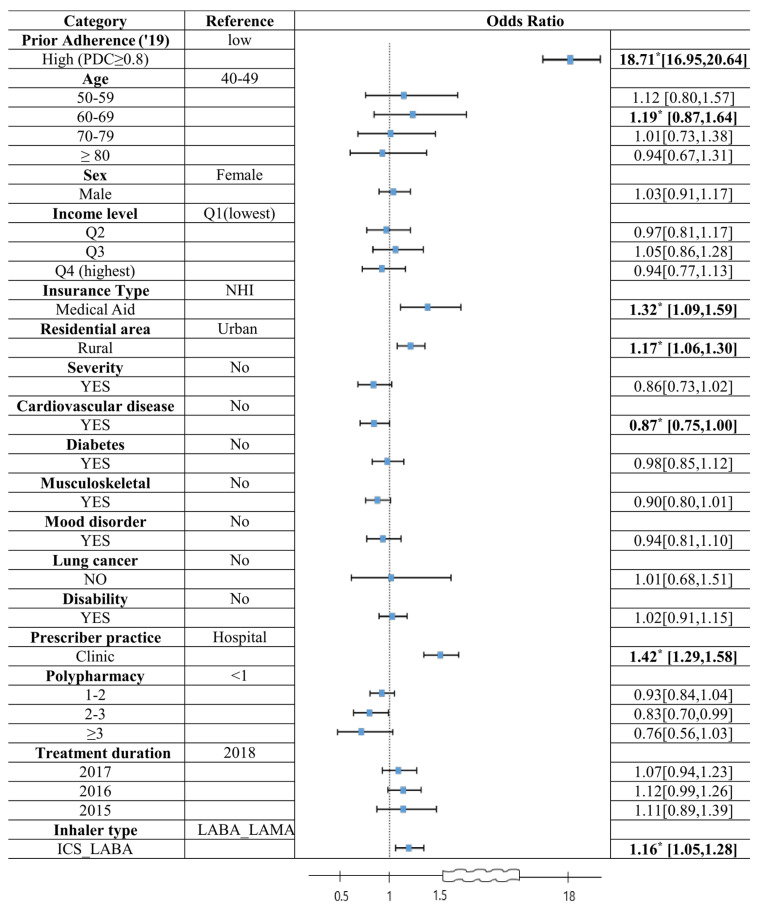
Multivariable logistic regression of factors associated with inhaler adherence during the COVID-19 lockdown. Note: LABA_LAMA, long-acting β2-agonists with long-acting muscarinic antagonists; ICS_LABA, inhaled corticosteroids with long-acting β2-agonists; * *p*-value under 0.05.

**Table 1 healthcare-13-01431-t001:** Major baseline characteristics of the study cohort and changes in inhaler adherence before and during the COVID-19 lockdown.

Category	Distribution		2019 PDC		2020 PDC		*p*-Value *
			Mean	SD	Mean	SD	
Total, N	15,971		56.3%	0.24	49.8%	0.27	<0.0001
Age, n (%)							
40–49	427	2.67%	54.8%	0.24	50.0%	0.27	<0.0001
50–59	1829	11.45%	56.8%	0.24	51.7%	0.27	
60–69	4642	29.07%	57.3%	0.24	51.9%	0.27	
70–79	6473	40.53%	56.5%	0.24	49.9%	0.27	
≥80	2600	16.28%	53.7%	0.25	45.1%	0.29	
Sex, n (%)							
Female	3316	20.76%	55.2%	0.25	48.7%	0.28	<0.0001
Male	12,655	79.24%	56.6%	0.24	50.2%	0.27	
Insurance type, n (%)							
NHI	12,558	78.63%	55.3%	0.24	49.0%	0.26	<0.0001
Medical Aid	3413	21.37%	59.9%	0.26	53.2%	0.29	
Residential area, n (%)							
Urban	9170	57.42%	56.0%	0.24	49.5%	0.27	<0.0001
Rural	6801	42.58%	56.7%	0.24	50.3%	0.27	
Cardiovascular disease, n (%)							
NO	13,293	83.23%	56.9%	0.24	50.6%	0.27	<0.0001
YES	2678	16.77%	53.0%	0.24	46.1%	0.26	
Prescriber practice setting, n (%)							
Hospital	8065	50.5%	54.6%	0.23	47.8%	0.26	<0.0001
Clinic	7906	49.5%	58.0%	0.25	51.9%	0.28	
Polypharmacy, n (%)							
<1	5329	33.37%	58.2%	0.25	51.5%	0.28	<0.0001
1~2	8094	50.68%	56.0%	0.24	49.7%	0.27	
2~3	1947	12.19%	53.6%	0.23	47.8%	0.26	
≥3	601	3.76%	50.5%	0.24	44.4%	0.27	
Medication treatment duration, n (%)							
2018	4659	29.17%	53.6%	0.26	46.9%	0.28	<0.0001
2017	4430	27.74%	54.5%	0.24	48.7%	0.27	
2016	6046	37.86%	58.7%	0.23	52.1%	0.26	
2015	836	5.23%	63.2%	0.23	56.0%	0.25	
Inhaler type, n (%)							
LABA_LAMA	7146	44.74%	55.1%	0.24	48.5%	0.27	<0.0001
ICS_LABA	8825	55.26%	57.2%	0.24	50.9%	0.27	

Note: PDC, proportion of days covered; LABA_LAMA, long-acting β2-agonists with long-acting muscarinic antagonists; ICS_LABA, inhaled corticosteroids with long-acting β2-agonists; * *p*-value under 0.05.

**Table 2 healthcare-13-01431-t002:** Comparison of major baseline characteristics between patients with high and low adherence during the COVID-19 lockdown.

Total, N	15,971		High Adherence		Low Adherence		*p*-Value *
			2880		13,091		
	N	%	N	%	N	%	
**Prior Adherence (’19), n (%)**							<0.0001
High adherence	3507	21.96%	2050	71.2%	1457	11.1%	
Low adherence	12,464	78.04%	830	28.8%	11,634	88.9%	
Age, n (%)							<0.0001
40–49	427	2.67%	74	2.6%	353	2.7%	
50–59	1829	11.45%	351	12.2%	1478	11.3%	
60–69	4642	29.07%	930	32.3%	3712	28.4%	
70–79	6473	40.53%	1124	39.0%	5349	40.9%	
≥80	2600	16.28%	401	13.9%	2199	16.8%	
Insurance type, n (%)							<0.0001
NHI	12,558	78.63%	2076	72.1%	10,482	80.1%	
Medical Aid	3413	21.37%	804	27.9%	2609	19.9%	
Residential area, n (%)							0.0165
Urban	9170	57.42%	1596	55.4%	7574	57.9%	
Rural	6801	42.58%	1284	44.6%	5517	42.1%	
Cardiovascular disease, n (%)							<0.0001
NO	13,293	83.23%	2510	87.2%	10,783	82.4%	
YES	2678	16.77%	370	12.8%	2308	17.6%	
Musculoskeletal, n (%)							0.0439
NO	11,479	71.87%	2114	73.4%	9365	71.5%	
YES	4492	28.13%	766	26.6%	3726	28.5%	
Prescriber practice setting, n (%)							<0.0001
Hospital	8065	50.50%	1186	41.2%	6879	52.5%	
Clinic	7906	49.50%	1694	58.8%	6212	47.5%	
Polypharmacy, n (%)							<0.0001
<1	5329	33.37%	1110	38.5%	4219	32.2%	
1~2	8094	50.68%	1415	49.1%	6679	51.0%	
2~3	1947	12.19%	283	9.8%	1664	12.7%	
≥3	601	3.76%	72	2.5%	529	4.0%	
Medication treatment duration, n (%)							<0.0001
2018	4659	29.17%	766	26.6%	3893	29.7%	
2017	4430	27.74%	753	26.1%	3677	28.1%	
2016	6046	37.86%	1171	40.7%	4875	37.2%	
2015	836	5.23%	190	6.6%	646	4.9%	
Inhaler type, n (%)							<0.0001
LABA_LAMA	7146	44.74%	1193	41.4%	5953	45.5%	
ICS_LABA	8825	55.26%	1687	58.6%	7138	54.5%	

Note: LABA_LAMA, long-acting β2-agonists with long-acting muscarinic antagonists; ICS_LABA, inhaled corticosteroids with long-acting β2-agonists; * *p*-value under 0.05.

## Data Availability

Data may be obtained from a third party and are not publicly available.

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
