# Peer review of "Impact of the COVID-19 Lockdown on Inhaler Adherence in Patients with COPD: A South Korean Nationwide Cohort Study"

_healthcare, 2025, doi:10.3390/healthcare13121431_

Round 1

Reviewer 1 Report

Comments and Suggestions for Authors

The manuscript presents an interesting and important topic in the field of COPD. The design and methodology are well-described, and the data analysis is rigorous and reliable. The discussion of the results is thorough and provides valuable insights into the potential implications of the research.

While manuscript is well-written, some minor revisions that includes the CAT score (if already conducted) could improve its clarity and impact. The discussion section could be expanded to provide more details on the implications of the findings.

Author Response

Thank you for your valuable comments. We have made the requested revisions as attached. Please refer to the revised manuscript for details.

Reviewer 2 Report

Comments and Suggestions for Authors

Dear authors,

Thank you for your submission and contribution.

This is an important study that aligns well with sustainable development goals related to health and well-being. 

Overall, the study is well-designed and well-implemented. The manuscript is well-written and informative. I have some review comments, recommendations, and suggestions for your consideration as follows.

Abstract

Background/Objectives: please add what COPD abbreviation stands for as this is the first time it appears in text. 

Keywords: it is recommended to arrange keywords in an alphabetical order. 

Materials and Methods

Study population: please define exclusion criteria as well for those excluded from the study.

Data Analysis

Please add how sample size was determined for the purpose of this study? were any sample size calculations conducted? 

Discussion

Limitations: in the current study, over 79% were male. How do you think this affected the results? would that be considered as a limitation? how did that affect internal and external validity and generalizability of the study results. Any recommendations to overcome such limitation for future direction and potential follow-up exploratory research studies? please elaborate. Also, the current study did not explore whether those patients with COPD lived with other family members and/or caregivers and how that would have affected adherence to medication and medication management? this could be explored in a future study to further support appropriate medication management for this population. 

I will be happy to look at the revised version.

Best wishes  

Author Response

(The authors gave the same response as above.)

Reviewer 3 Report

Comments and Suggestions for Authors

Thank you for inviting me to review this manuscript on hot topic!

I have several proposals:

1, the tables are very huge please revise them with possibility to transfer in appendix!

2, discussions must be extended, especially I propose to analyse the situation in the geographical region

3. please elaborate future directions chapter

4. please highlight what instruments were used for assessment of adherence and what new possibilities for assessment of adherence can be proposed (here you can use data from ENABLE COST project)

5. Please analyse more what the real impact of low adherence in COPD in your region, how this can influence real life practice and clinical local guidelines! 

Author Response

(The authors gave the same response as above.)

Reviewer 4 Report

Comments and Suggestions for Authors

1. Introduction
- Please clarify why South Korea was chosen as the location for this study. What distinguishes this country from other Asian nations during the COVID-19 pandemic?
- Discuss the continued relevance of this study, even though the COVID-19 crisis has largely subsided. Highlight its importance not only for policymakers but also for the broader community.
- Although many studies have explored COVID-19-related research, please explain the novel aspects of this study!

2. Materials and Methods
- In the sentence, "To identify factors associated with high adherence (PDC ≥80%) during the COVID-19 lockdown, we performed multivariable logistic regression analysis," please specify the reference base used for the high adherence standard (PDC ≥80%). This information should be included in the manuscript.
- Are there any inclusion and exclusion criteria for selecting participants in this study? Please outline these criteria in the Materials and Methods section.

3. Results
- Table 1 shows that for TOTAL, N has a p-value <0.0001. Please clarify what this p-value indicates regarding TOTAL N.
- In Table 2, why do the total percentages not add up to 100% for Polypharmacy (n, %) and Medication Treatment Duration (n, %) categories?

4. Conclusion
The conclusion should reinforce the main points of the study and emphasize its practical relevance. Instead of merely restating previous sections, use this section to propose concrete next steps for health systems or policymakers.

5. References
- Please ensure that the references are formatted correctly according to the journal's guidelines.

Author Response

(The authors gave the same response as above.)

Round 2

Reviewer 3 Report

Comments and Suggestions for Authors

accept as it